# Who supports science-related populism? A nationally representative survey on the prevalence and explanatory factors of populist attitudes toward science in Switzerland

**Niels G. Mede**[1]*, **Mike S. Schäfer**[1], **Julia Metag**[2], **Kira Klinger**[2]

**1** University of Zurich, Zurich, Switzerland, **2** University of Münster, Münster, Germany

* n.mede@ikmz.uzh.ch

**Data Availability Statement:** The R code we used for the statistical analyses is available at https://osf.io/qj4xr/. Survey data and additional materials (e.g., the questionnaires and a methodological report,

## Abstract

Science and its epistemology have been challenged by *science-related populism*—a variant of populism suggesting that a virtuous "ordinary people," and not allegedly corrupt academic elites, should determine the "production of truth." Yet almost no studies have assessed the prevalence of *science-related populist attitudes* among the population and explanatory factors thereof. Based on a nationally representative survey in Switzerland, our study shows that only a minority of the Swiss exhibit science-related populist attitudes. Comparisons with reference studies suggest that these attitudes may be less prevalent in Switzerland than political populist attitudes. Those who hold stronger science-related populist attitudes tend to have no university education, less personal contact with science, lower scientific literacy, and higher interest in science. Additional analyses show that left-leaning citizens are less likely to hold science-related populist attitudes than moderate and right-leaning citizens. Our findings contribute to current debates about a potential fragmentation of science communication audiences and call for further research on the sociodemographic and attitudinal profiles of people with skeptical orientations toward science.

## Introduction

Various criticisms of scientific expertise have recently been circulating in public. Some of them show distrust toward scientific knowledge on climate change or the COVID-19 pandemic [1] and valorize commonsensical epistemologies or "experiential folk wisdom" [2]. Journalists and scholars have thus diagnosed a "breach of faith with science" [3] and portrayed (particularly Western) societies as being polarized between those who endorse scientific expertise and value rationalism, and those who reject scientific expertise and rely on intuition [4]. Empirical investigations provide preliminary evidence for this, showing that certain anti-scientific views indeed prevail in specific segments of the population [5].

However, these diagnoses and investigations only focused on isolated aspects of public criticisms of science and did not offer integrated conceptual and methodological approaches.

the former in German, French, and Italian, the latter only in German) are publicly available in the online repository SWISSUbase (doi: 10.48573/wpf5-hf36).

**Funding:** This research received financial support by the Gebert Rüf Foundation (https://www.grstiftung.ch/en.html), the Mercator Foundation Switzerland (https://www.stiftung-mercator.ch/), and the University of Zurich (https://www.uzh.ch/cmsssl/en.html). Funding was acquired by J.M. and M.S.S. The funders had no role in study design, data collection and analysis, decision to publish, or preparation of the manuscript.

**Competing interests:** The authors have declared that no competing interests exist.

Mede and Schäfer [6] proposed one such approach by conceptualizing *science-related populism*, which describes a perceived antagonism between an allegedly virtuous "ordinary people" and allegedly immoral scientists, experts, and other members of "academic elites" [6]. This antagonism manifests in claims demanding that the commonsensical and experiential knowledge of the people should be superior to the allegedly useless and ideologically biased knowledge of elites. In addition, Mede, Schäfer, and Füchslin [7] introduced the "SciPop Scale," a robust and reliable survey scale to measure *science-related populist attitudes* in empirical research.

Existing scholarship suggests that some segments of many (Western) populations hold *science-related populist attitudes* [8, 9]. However, most of this scholarship did not rely on the concept of science-related populism or employ the SciPop Scale to measure it, and focused only on single facets of science-related populist attitudes. Accordingly, there is no reliable evidence on which parts of the public support science-related populist attitudes, how such support is linked to other attitudes, or whether concerns of societal polarization along populist attitudes toward science are warranted. This study provides such evidence, relying on a nationally representative survey in Switzerland—a country that exhibits conditions that potentially facilitate science-related populism: On the one hand, Switzerland has been assumed to offer fertile ground for populism in general, due to influential populist parties like the Swiss People's Party (SVP) [10] and a political system that grants much power to the people [11], and for anti-scientific ideas in particular, due to the pervasiveness of certain pseudoscience claims within public discourse [12]. On the other hand, however, a notable density of high-quality scientific institutions, high public trust in science, and a highly educated citizenry may render Switzerland resilient to science-related populism [13].

We examined the following research questions:

RQ1: *How prevalent are science-related populist attitudes among the Swiss population*?

RQ2: *How are these attitudes related to sociodemographic characteristics, political orientation and religiosity, and general perceptions of science*?

## Conceptual framework and existing research

### Science-related populism

Populism in general describes societies as being divided into two antagonistic groups: "the people" and "the elite" [14]. "The people" are conceived as a collective of virtuous citizens who supposedly have similar needs, shared values, and a common will which ought to be the exclusive reference point of political decision-making. Meanwhile, "the elite" is seen as a minority group of powerful authorities who lead society, yet (allegedly) ignore the people's legitimate will [15].

Populist ideas often pit the people against *political* elites. But they may also target *academic elites*, criticizing scientific instead of political authorities and discussing scientific truth claims instead of political power claims [16]. Such anti-scientific ideas represent a variant of populism that differs conceptually from political populism—and has been conceptualized as "science-related populism" [6].

Science-related populism assumes an antagonism between the "ordinary people" and the "academic elite" that is allegedly due to the elite depriving the people of "decision-making sovereignty" and "truth-speaking sovereignty" [6]. Therefore, it involves four conceptual components:

1. *The ordinary people*, who are conceived as a homogenous group of citizens who consider seemingly authentic common sense, everyday experiences, and gut feelings as "true knowledge" and leading decision-making principles [17];

2. *The academic elite*, which is portrayed as a collective of immoral scientists and experts who allegedly ignore the people's needs and intrude on their commonsensical truths and decisions with seemingly useless and ideologically biased scientific knowledge [9];

3. *Decision-making sovereignty*, which refers to the right to determine research agendas or funding expenditures—a right academic elites allegedly claim for themselves, although, according to populists, it should be with the people [18];

4. *Truth-speaking sovereignty*, which refers to the right to define "true knowledge"—a right academic elites allegedly also deny the people [19].

Importantly, these four components are non-compensatory, which means that their concurrent presence is required to diagnose science-related populism [20]. If, for example, someone only claims that their knowledge is superior to that of scientists, but otherwise does not demand a say in decisions about the allocation of research funding, then this may qualify as anti-intellectualism rather than "full-fledged" science-related populism [9], because demands for decision-making sovereignty and truth-speaking sovereignty are both prerequisites of science-related populism [7].

The non-compensatoriness of the components of science-related populism has conceptual and methodological implications. On the one hand, it suggests that science-related populism is conceptually different from other anti-scientific phenomena: (Dis)trust in science, for example, mainly describes perceptions of scientific knowledge, institutions, and experts (i.e., attitudes toward the academic elite), yet it does not necessarily include people's demands for participation in the production and definition of legitimate knowledge (i.e., claims for decision-making and truth-speaking sovereignty) [21]. Trust thus overlaps with science-related populism but lacks two crucial components of it—which manifests in only moderate statistical correlations between the two concepts, as representative survey research shows [7]. On the other hand, non-compensatory concepts like science-related populism require specific methodological approaches—such as the "Goertz approach," which prevents erroneously assuming the presence of science-related populism when only isolated components are being observed (see Data and Method section) [22].

## Reviewing the scholarly literature: What do we know about science-related populist attitudes and their prevalence in the population?

Science-related populism can manifest in individuals' orientations toward science and expertise, i.e., as *science-related populist attitudes* [6]. These attitudes have been conceptualized as a four-dimensional construct and were operationalized with a robust and reliable survey scale [7], but have not been systematically investigated in empirical research. However, a number of studies analyzed isolated components of science-related populist attitudes (but often used different labels), focusing on at least one of their theoretical dimensions—i.e., on individuals' *conceptions of the ordinary people*, *conceptions of the academic elite*, *demands for (science-related) decision-making sovereignty*, or *demands for truth-speaking sovereignty* [7].

First, favorable *conceptions of the ordinary people* have been found in survey research indicating that sizeable portions of Western populations perceive ordinary people as virtuous and homogeneous [23]. Some of these conceptions also address *epistemology-related* virtuousness and homogeneity perceptions: While 48% of Americans agree that "ordinary people are perfectly capable of deciding for themselves what is true and what is not" [24], 41% of Austrians think that relying on common sense is what collectivizes these people [25].

Second, unfavorable *conceptions of the academic elite* seem less widespread, but still appear in some countries: Representative surveys show that 14% of Italians agree that "people with

advanced degrees do not understand the problems of ordinary people" [26] and 42% of Belgians believe that "people who have studied for a long time and have many diplomas do not really know what makes the world go round" [27]. Similar results have been found for specific science-related topics: Representative surveys on public views of the COVID-19 pandemic have shown, for example, that considerable shares of the Polish, French, Spanish, Italian, and Portuguese public assume that "there's too much experts don't tell us" [8]. Qualitative research suggests that some of these resentments are rooted in beliefs that scientists apply "fuzzy and messy" practices [18], follow "hidden motives and agendas" [28], and conspire with other elites [2].

Third, people's *demands for (science-related) decision-making sovereignty* have been observed in surveys showing that 19% of the Swiss want to have a say in "decisions about the topics scientists research," for example [29]. Claims suggesting that people's experiential knowledge should have more influence on political decisions are much more widespread: 77% of Germans think that solving their problems requires "people with practical experience" rather than experts without such experience [5]. Qualitative studies exploring more radical variants of such views describe how conspiracy theory advocates maintain that an academic elite denies ordinary people empowerment within the process of knowledge production [18].

Fourth, people's *demands for truth-speaking sovereignty* have been found in segments of Western publics: 44% of Americans claim that experts "ought to be less trusted than ordinary people" [30], 33% of Germans demand that people should rely more on common sense and less on scientific studies [31], and 12% of Italians and 15% of Romanians strongly agree that one should rather "trust in the wisdom of ordinary people than the opinions of experts" [re-analysis of 32's replication data]. Some members of these segments may also consider "maternal instinct" [33] and "subjective feelings" [18] as legitimate truths, as qualitative interview studies demonstrate.

Overall, scholarship indicates that certain milieus in several (Western) countries, including Switzerland, are more inclined than others to endorse aspects of science-related populism, which supports the assumption that some of them may also support the full spectrum of science-related populist attitudes. This assumption, however, remains to be scrutinized empirically—which we do in this study.

## Science-related populist attitudes and explanatory factors

Scholarship suggests that endorsing science-related populist attitudes is not idiosyncratic: Research on similar phenomena—although almost only for Western and primarily for US populations—has shown that critical attitudes toward scientific expertise often depend on people's personal background, their political or religious views, and other orientations toward science, such as (dis)trust in scientists [34, 35]. Science-related populist attitudes may therefore be explained to some extent by people's *sociodemographic characteristics*, *political orientation and religiosity*, and *general perceptions of science*.

Yet existing research is limited in that it has not addressed the full range of science-related populist attitudes. Thus, it allows only for tentative assumptions about explanatory factors—but it still provides worthwhile starting points for exploring such factors.

**Sociodemographic characteristics.**   Scholarship suggests that older men with low formal education and little personal contact with science are more likely to harbor populist attitudes toward science [6, 36, 37]. Members of this milieu could be understood as "losers of automation" who fear that scientific and technological innovations threaten their socioeconomic position [38], worry that certain sciences and humanities erode cultural values [39], and avoid personal relationships with scientists as they see them as a distant elite [19]. Some scholars

maintain that these people live in rural regions [40], whereas others emphasize the "urban roots" [41] of populist phenomena—assumptions which touch upon the notion of a center-periphery cleavage dividing modern societies [42].

But empirical evidence for these arguments is inconclusive. On the one hand, several studies have shown that men, older adults, people with lower formal education, or inhabitants of rural regions are indeed more likely to support political populism [43], lack confidence in science [9], consider scientific innovations like artificial intelligence as harmful for society [5], have anti-intellectual attitudes [44], and think that "common wisdom" [45] and the "opinion of ordinary people" [27] supersede expert knowledge [re-analysis of 46's replication data]. Yet on the other hand, critical orientations toward science and political populist attitudes were also found among women [34] and people who are younger [47], have better formal education [re-analysis of 46's replication data], and reside in big cities [48].

Existing findings are also ambiguous as to how prevalent (political) populist attitudes are in the different linguistic regions of Switzerland: Polls and elections demonstrate pronounced discrepancies in political and science-related attitudes between the German-, French-, and Italian-speaking regions [13, 49], which have been attributed to cultural differences across these regions [50]. But while some studies suggest that these differences made populism more likely to thrive in French-speaking regions [51], others indicate the opposite [10].

Overall, scholarship is not entirely conclusive about how sociodemographic factors are associated with (phenomena akin to) science-related populism. One reason may be that most studies focus on specific manifestations of political populism (e.g., left-wing vs. right-wing populism) or anti-scientific sentiment (e.g., criticism of epistemic authorities vs. denial of scientific knowledge), both of which have often been described as rather heterogeneous and context-dependent [43, 52]. It is thus unclear how science-related populist attitudes vary across sociodemographic population groups (see RQ2). But that they do vary somehow is reasonable to assume, so we hypothesize:

H1: *Science-related populist attitudes are associated with* (a) *age*, (b) *gender*, (c) *formal education*, (d) *proximity to science*, (e) *urbanity*, *and* (f) *region of residence*.

**Political orientations and religiosity.**　Populist demands toward science may thrive in discourses around politicized scientific topics, i.e., topics whose perceptions are linked to ideological agendas and sometimes drastic political claims (e.g., climate change) [6]. Science-related populists may thus tend to subscribe to extreme political views [53]. However, it is not clear if they then identify more with left- or right-leaning views—or, similar to many voters of certain populist parties, with none of these views [54–56]. Conceptually, science-related populism is not tied to either side, because it reflects fundamental beliefs about science which may occur across the political spectrum. However, some have suggested that populist resentment toward science is prone to align with right-leaning political ideologies [39]. This argument is supported by empirical research showing that low trust in scientists, hostile attitudes toward scientists, mistrust of experts, and preferences for commonsensical knowledge are associated more with right-leaning or conservative views [5, 34, 45, 57]. Yet others argued that critical attitudes toward certain scientific disciplines are more pronounced among left-leaning voters [58]. Further scholarship indicates that examining public beliefs about science requires discerning different nuances of political orientations [59].

The scientific epistemology does not only conflict with populist appeals to the common sense of "ordinary people," but also with religious doctrines [60]. Science-related populist attitudes may thus coincide with higher levels of religiosity. Research on anti-intellectualism [9] and distrust toward scientists [48] supports such a correlation—but multiple studies indicate that it only occurs for certain religions [61], geographical contexts [62], scientific topics [63], and variants of science attitudes [64].

Accordingly, we expect science-related populist attitudes to be associated with religious and political views, but refrain from hypothesizing about the direction of such associations:

H2: *Science-related populist attitudes are associated with* (a) *political orientation and* (b) *religiosity*.

**General perceptions of science.** Science-related populist attitudes are probably also associated with general perceptions of science: *Interest in science* and *scientific literacy*, for example, could be lower among science-related populists, because they tend to reject established science and may thus be less interested in it and reluctant to learn about it [65]. Nevertheless, populist ideation about science and its epistemology requires a certain level of cognitive engagement with questions about the role of scientific knowledge in society and people's daily life. Consequently, science-related populism could also coincide with higher levels of interest in science and scientific literacy. Empirical evidence is not conclusive: Studies indicate that low interest in science may correlate with resentment toward it—but they often focused on specific topics such as climate change [66, 67]. And while some studies show that scientific literacy is lower among people with pseudoscientific beliefs [68] and low confidence in scientists [69], others do not find a link between negative attitudes toward science and knowledge about it [47], or suggest that links depend on the topical context [52].

Scholarship further indicates that science-related populist attitudes are connected to *trust in science* and *trust in scientists*: It assumed that trust can reduce negative attitudes toward science, because trustworthiness perceptions partly derive from a rather stable "propensity to trust" [21] that can attenuate more volatile reservations against science—such as science-related populist attitudes, which may fluctuate considerably over time [70]. However, the reverse causality is also conceivable, as trust can be as volatile as science-related populist attitudes and may therefore be reduced by them [53, 65, 72]. Empirical research offers little evidence on the direction of the relationship between trust in science and populist attitudes toward it, but it does indicate that such a relationship exists: For example, it has been found that trust in science and scientists tends to be lower among people who hold pseudoscientific beliefs [71], refrain from deferring to scientific authority [72], and believe that science should accept other forms of knowledge like common sense [73]. However, trust in science can also translate into a blind faith that leaves people more prone to pseudoscience, so it is not entirely clear if trust and science-related populism are negatively or positively associated [74]. Therefore, we hypothesize:

H3: *Science-related populist attitudes are associated with* (a) *interest in science*, (b) *scientific literacy*, (c) *trust in science, and* (d) *trust in scientists*.

## Data and method

### Data

We examined the research questions and hypotheses in a nationally representative survey in Switzerland—the "Wissenschaftsbarometer Schweiz" ($N$ = 1,050; age: $M$ = 48.3 years; $SD$ = 17.3; gender: 53.5% female; education: 47.8% university degree). Data were collected by the polling company *DemoSCOPE* from 17 June to 20 July 2019 in computer-assisted telephone interviews (landline: 81%; mobile: 19%) with respondents from the three Swiss linguistic regions (German-, French-, and Italian-speaking). Landline respondents were contacted via numbers from public telephone listings and were selected based on gender and age quotas, while mobile respondents were recruited via random digit dialing. 2.6% of all calls resulted in completed interviews, and 21.7% were answered but interview requests were declined. 75.7%

were not picked up or reached a dead number. In the analyses, the data were weighted regarding linguistic region, gender, age, and the probability that respondents were reached via landline or mobile (weights: $M = 1.00$, $SD = 1.44$, range = 0.04 to 23.56).

## Measures

*Science-related attitudes* were measured with the SciPop Scale, a survey scale validated in representative German-, French-, and Italian-speaking samples of the Swiss population [7]. It consists of eight items capturing the four conceptual components of science-related populist attitudes with 2-item subscales (see Table A1 in S1 Appendix for an overview of all variables used in the analyses). Scale items ask respondents, for example, if they agree that "ordinary people are of good and honest character" (conceptions of the ordinary people), "scientists are only after their own advantage" (conceptions of the academic elite), "the people should have influence on the work of scientists" (demands for decision-making sovereignty), and one "should rely more on common sense and less on scientific studies" (demands for truth-speaking sovereignty). Agreement was measured with 5-point Likert scales (1 = "fully disagree"; 5 = "fully agree").

To obtain a single aggregate score which quantifies propensity and aversion to science-related populism, we followed the "Goertz approach" [22]: We computed mean values of each of the four 2-item subscales for every respondent and determined the smallest of these four values to represent their "SciPop Score." The SciPop Score thus ranges from 1.00 (full disagreement with both items of at least one subscale) to 5.00 (full agreement with all items).

We applied the Goertz approach because it operationalizes the theoretical premise that science-related populism is a non-compensatory concept, i.e., that it relies on the concurrent presence of all its four components (see S1 Appendix for more details) [7]. Other aggregation approaches—like the "Bollen approach", which averages all scale items or uses factor scores of confirmatory factor analysis, or the "Sartori approach", which classifies respondents as populist vs. non-populist based on their responses—do not account for the concurrency premise [22]. They are thus less useful to capture science-related populism or other populism variants [75], which is why recent research on political populism increasingly favors Goertzian over Bollenian or Sartorian aggregation procedures [76–78].

However, the four theoretical components of science-related populism themselves do not represent non-compensatory concepts. This means that they must be operationalized—most usefully—with the Bollen approach, which is common practice in current populism research [79–81]. Accordingly, we used respondents' average agreement with the four subscales of the SciPop Scale to quantify *conceptions of the ordinary people*, *conceptions of the academic elite*, *demands for decision-making sovereignty*, and *demands for truth-speaking sovereignty* (possible range of each subscale score: 1.00 to 5.00). Overall, higher SciPop Scores and higher subscale scores indicate stronger support for science-related populism and its components.

Explanatory variables were measured as follows: *Age* (in years) and *formal education* (three levels) were reported by the respondents, *gender* was identified by the interviewer. *Proximity to science* was measured with a score composed of four dichotomous items asking respondents whether they are scientists themselves, know a scientist personally, work with scientists, and have family members who are or have been studying [29]. *Urbanity* of respondents' residence places was operationalized with log-transformed inhabitant counts of their residence municipalities. Moreover, we included a categorical variable indicating the *region of residence* (German-, French-, or Italian-speaking). *Political orientation* was measured with a 7-point Likert scale (1 = "very left-leaning", 7 = "very right-leaning"), while 5-point Likert scales were used to measure *religiosity* (1 = "not at all religious", 5 = "very religious"), *interest in science* (1 = "not

interested at all", 5 = "very strongly interested"), *trust in science*, and *trust in scientists* (both 1 = "very low", 5 = "very high"). *Scientific literacy* was measured by asking respondents to assess if five statements on factual and procedural science knowledge were correct [82].

Respondents could also answer "don't know" or refuse to answer to questions, including the political orientation item measuring left-right self-identification. Running additional analyses that included these respondents allowed us to test if science-related populists—similar to political populists [54–56]—are less willing or able to position themselves on the left-right spectrum of political orientations.

## Analytical comparison with reference studies

We added further perspective to this study by comparing our findings on *science-related* populist attitudes with those from recent survey research on *political* populist attitudes. We reviewed four representative survey studies which analyzed political populism in Switzerland and Germany and extracted the *political populism scores* they report [51, 76, 83, 84]. To obtain equivalent *science-related populism scores* for our respondents, we replicated the computational procedures of the reference studies with our data: Like Rico and Anduiza [83], we computed mean values of all SciPop Scale items, which yielded a Bollenian SciPop Score. Like van Hauwaert et al. [51], we extracted factor scores of a polychoric confirmatory factor analysis with all SciPop Scale items, which yielded another Bollenian SciPop Score. Like Vehrkamp and Merkel [84], we applied a three-categorical classification scheme, which yielded a Sartorian SciPop Score. Stier et al. [76] followed the Goertz approach as well, so we used the Goertzian SciPop Scores from our main analysis for comparison with their results.

## Results

### Prevalence of science-related populism

Descriptive analyses offer two main insights into the prevalence of science-related populist attitudes in Switzerland (RQ1). First, we find relatively low levels of science-related populism in the Swiss population: Only 2.8% endorse all four components (i.e., reach SciPop Scores between 4.00 and 5.00). In turn, a majority of 55.1% of the Swiss reject at least one component (i.e., reach SciPop Scores between 1.00 and 2.00), which results in an overall average SciPop Score of 2.22 ($SD$ = 0.80).

Comparisons with the four reference studies on political populism in Switzerland and its neighboring country Germany suggest that *science-related* populist attitudes are not widespread in Switzerland—at least less widespread than *political* populist attitudes (see Table 1): Stier et al. [76], who investigate Germany, find an average Goertzian political populism score of 2.98 ($SD$ = 0.85). Similarly, Rico and Anduiza [83] and Vehrkamp and Merkel [84] observe political populism levels higher than the science-related populism level we estimate when replicating their analytical procedures. Only van Hauwaert et al. [51] report a populism score lower than ours.

Second, our analyses show that parts of the Swiss population nevertheless endorse components of science-related populism, and that endorsement differs across components: On the one hand, Respondents tend to be more inclined to hold favorable *conceptions of the ordinary people*, which suggest that they are virtuous and united ($M$ = 3.29, $SD$ = 0.92; see Table A2 in S1 Appendix), and to have populist *demands for truth-speaking sovereignty*, which claim that commonsensical knowledge should be superior to scientific knowledge ($M$ = 3.16, $SD$ = 0.96). On the other hand, *demands for decision-making sovereignty* calling for more popular influence on scientists' work ($M$ = 2.87, $SD$ = 0.97) and unfavorable *conceptions of the academic elite*

**Table 1. Comparison of science-related populism scores (present study) and political populism scores (reference studies).**

| Populism score computation | Science-related populist attitudes (present study) | | | Political populist attitudes (reference study) | | | | | | | |
|---|---|---|---|---|---|---|---|---|---|---|---|
| | M | SD | | Reference study | Populism scale used | M | SD | | Country | Year | N |
| Minimum subscale mean (Goertz approach) | 2.22 | 0.80 | | [76] | [110] | 2.98 [a] | 0.85 [a] | | Germany | 2019 | 979 |
| Mean across all items (Bollen approach) | 3.02 [b] | 0.66 | | [83] | [111] | 3.62[c] | n.a. | | Switzerland | 2015 | 2,046 |
| CFA scores[d] (Bollen approach) | 0.08 | 0.20 | | [51] | [111], new items | -0.30 [e] | 0.21 [e] | | Switzerland | 2015 | 2,046 |
| | Populist | Non-populist | Mixed | Reference study | Populism scale used | Populist | Non-populist | Mixed | Country | Year | N |
| Categorical classification[f] (Sartori approach) | 1.0% | 50.7% | 48.2% | [84] | [111, 112], new items | 20.9% | 47.1% | 32.0% | Germany | 2020 | 10,055 |

[a] These *M* and *SD* values are not reported in Stier et al. [76], but result from analyses we performed with their original data. In these analyses, we computed a Goertzian populism score in the same way we did in this current study (see Data and Method section).

[b] Cronbach's Alpha = 0.76 (95% CI [0.74, 0.78]).

[c] Rico and Anduiza [83] report a mean of *M* = 2.62 for Switzerland in the online appendix of their article. However, they used five-point Likert scales running from "strongly disagree" (coded 0) to "strongly agree" (coded 4), whereas we used five-point Likert scales running from "fully disagree" (coded 1) to "fully agree" (coded 5). To allow for comparison between our and their study, we added 1.00 to the mean they report.

[d] Replicating the approach of van Hauwaert et al. [51], we ran a polychoric confirmatory factor analysis (CFA) with all SciPop Scale items permitted to load on one latent factor, and used factor scores as individual SciPop estimates. CFA model fit was unsatisfying ($N = 986$; $\chi2 = 609.750$, $df = 20$, $p < 0.001$; CFI = 0.824, TLI = 0.753, RMSEA = 0.173, SRMR = 0.083), which is likely due to the non-compensatory nature of science-related populist attitudes [7].

[e] *M* and *SD* of average CFA scores per Swiss canton [51].

[f] Replicating the approach of Vehrkamp and Merkel [84], we applied the following coding to assign respondents to three categories: Only those who indicated agreement to all eight SciPop Scale items (i.e., reported a 4 or 5) were classified as "populist." Respondents who indicated complete disagreement with at least one item (i.e., reported a 1), or who indicated moderate disagreement with at least half of the items (i.e., reported a 2) were classified as "non-populist." All other respondents were classified as "mixed."

portraying it as corrupt and conspiring (*M* = 2.77, *SD* = 0.88) are slightly less pronounced in Switzerland (see t-tests in Table A3 in S1 Appendix).

## Explaining science-related populist attitudes

To investigate how science-related populist attitudes and their four components relate to people's sociodemographic characteristics, political orientation and religiosity, and general perceptions of science (RQ2), we followed four analytical steps.

First, we fitted a multiple linear regression model explaining science-related populist attitudes with all explanatory factors described in the Data and Method section. Results suggest that *only few sociodemographic characteristics affect affinity to science-related populism*: Its supporters and opponents do not differ substantially in age (H1a), gender (H1b), or their likelihood to reside in urban or rural areas (H1e; see Table 2). Overall, the full set of sociodemographic variables explains only 10.4% of the variance of science-related populist attitudes (*F (df)* = 7.72 (8, 906), $p < 0.001$; see stepwise regression results in Table A4 in S1 Appendix). The regression model also indicates that science-related populists are not generally more prone to prefer right-leaning over left-leaning political positions or vice versa (H2a; but see additional analyses below) or to be more or less religious than the rest of the population (H2b).

But science-related populist attitudes are nevertheless not randomly distributed in Switzerland. Science-related populists are less likely to hold a university degree (H1c), have less

**Table 2. Multiple linear regressions explaining science-related populist attitudes and its components.**

| Explanatory Variable | Science-related populist attitudes | | | Conceptions of the ordinary people | | | Conceptions of the academic elite | | | Demands for decision-making sovereignty | | | Demands for truth-speaking sovereignty | | |
|---|---|---|---|---|---|---|---|---|---|---|---|---|---|---|---|
| | b (SE) | β | p | b (SE) | β | p | b (SE) | β | p | b (SE) | β | p | b (SE) | β | p |
| Intercept | 2.89 (0.42) | 2.21 | < 0.001 | 3.78 (0.38) | 3.29 | < 0.001 | 4.21 (0.37) | 2.76 | < 0.001 | 2.90 (0.41) | 2.84 | < 0.001 | 4.59 (0.42) | 3.14 | < 0.001 |
| Age | 0.00 (0.00) | 0.00 | 0.967 | 0.01 (0.00) | 0.29 | 0.002 | 0.00 (0.00) | 0.12 | 0.204 | -0.01 (0.00) | -0.37 | < 0.001 | 0.01 (0.00) | 0.42 | < 0.001 |
| Gender (0 = male, 1 = female) | 0.07 (0.08) | 0.07 | 0.441 | -0.04 (0.08) | -0.04 | 0.660 | -0.06 (0.08) | -0.06 | 0.454 | 0.05 (0.09) | 0.05 | 0.571 | 0.03 (0.09) | 0.03 | 0.711 |
| Education (ref. secondary education) | | | | | | | | | | | | | | | |
| Compulsory school | -0.37 (0.15) | -0.37 | 0.012 | 0.15 (0.11) | 0.15 | 0.171 | -0.15 (0.12) | -0.15 | 0.183 | -0.31 (0.16) | -0.31 | 0.059 | -0.15 (0.13) | -0.15 | 0.272 |
| University degree | -0.16 (0.08) | -0.16 | 0.042 | -0.25 (0.10) | -0.25 | 0.010 | -0.02 (0.08) | -0.02 | 0.817 | -0.19 (0.10) | -0.19 | 0.064 | -0.28 (0.09) | -0.28 | 0.003 |
| Proximity to science | -0.11 (0.04) | -0.26 | 0.002 | -0.10 (0.04) | -0.23 | 0.008 | -0.09 (0.04) | -0.21 | 0.019 | -0.09 (0.04) | -0.21 | 0.044 | -0.13 (0.04) | -0.31 | 0.002 |
| Urbanity of residence | -0.01 (0.03) | -0.03 | 0.713 | -0.11 (0.03) | -0.32 | < 0.001 | -0.01 (0.02) | -0.04 | 0.585 | -0.04 (0.03) | -0.13 | 0.111 | -0.04 (0.03) | -0.13 | 0.156 |
| Swiss region (ref. French-speaking) | | | | | | | | | | | | | | | |
| German-speaking | -0.17 (0.11) | -0.17 | 0.106 | 0.11 (0.10) | 0.11 | 0.242 | -0.26 (0.09) | -0.26 | 0.007 | -0.03 (0.12) | -0.03 | 0.793 | 0.06 (0.10) | 0.06 | 0.563 |
| Italian-speaking | -0.24 (0.12) | -0.24 | 0.044 | 0.12 (0.12) | 0.12 | 0.289 | -0.39 (0.13) | -0.39 | 0.002 | -0.17 (0.18) | -0.17 | 0.344 | -0.21 (0.12) | -0.21 | 0.089 |
| Political orientation (1 = left, 7 = right) | 0.09 (0.05) | 0.24 | 0.052 | 0.06 (0.03) | 0.16 | 0.046 | 0.03 (0.04) | 0.08 | 0.427 | 0.11 (0.04) | 0.28 | 0.007 | 0.04 (0.04) | 0.12 | 0.305 |
| Religiosity | 0.02 (0.04) | 0.04 | 0.626 | 0.01 (0.04) | 0.03 | 0.729 | 0.06 (0.03) | 0.15 | 0.067 | 0.08 (0.04) | 0.19 | 0.058 | 0.04 (0.03) | 0.10 | 0.255 |
| Interest in science and research | 0.09 (0.03) | 0.18 | 0.013 | 0.04 (0.04) | 0.09 | 0.333 | 0.03 (0.04) | 0.06 | 0.484 | 0.17 (0.04) | 0.35 | < 0.001 | -0.03 (0.04) | -0.07 | 0.425 |
| Scientific literacy | -0.04 (0.01) | -0.27 | < 0.001 | -0.05 (0.01) | -0.36 | < 0.001 | -0.02 (0.01) | -0.10 | 0.197 | -0.04 (0.01) | -0.24 | 0.007 | -0.03 (0.01) | -0.22 | 0.012 |
| Trust in science | -0.05 (0.08) | -0.08 | 0.501 | 0.13 (0.09) | 0.20 | 0.122 | -0.12 (0.08) | -0.18 | 0.152 | 0.06 (0.08) | 0.09 | 0.421 | -0.13 (0.08) | -0.19 | 0.109 |
| Trust in scientists | -0.12 (0.07) | -0.19 | 0.098 | -0.10 (0.06) | -0.16 | 0.108 | -0.25 (0.07) | -0.39 | < 0.001 | -0.04 (0.08) | -0.06 | 0.622 | -0.23 (0.07) | -0.35 | 0.002 |
| *Adj. R²* | 0.18 | | | 0.21 | | | 0.18 | | | 0.11 | | | 0.30 | | |
| *F (df)* | 10.28 (14, 900) | | < 0.001 | 10.21 (14, 903) | | < 0.001 | 6.43 (14, 910) | | < 0.001 | 4.50 (14, 910) | | < 0.001 | 16.76 (14, 908) | | < 0.001 |
| *AIC* | 2434.00 | | | 2670.44 | | | 2588.77 | | | 2895.30 | | | 2676.75 | | |
| *N* | 915 | | | 918 | | | 925 | | | 925 | | | 923 | | |

*Note*: Values indicated are standardized regression coefficients. Regressions were run with survey weights using the R package survey v4.1–1 [113]. Standardization of b coefficients follows Gelman's [114] suggestion to rescale the estimates by dividing them by two standard deviations instead of one. Assumption checks, which can be reproduced with the R syntax, neither suggest multicollinearity of explanatory variables nor non-normality or heteroskedasticity of the residuals of any of the regression models.

proximity to science (H1d), and tend not to live in the Italian-speaking region of Switzerland (H1f; but see additional analyses below). They also seem to hold distinctive general perceptions of science, having more interest in it (H3a) but considerably lower scientific literacy than people averse to science-related populism (H3b). However, we do not find that science-related

populism and trust in science or scientists have a relationship that is independent of the socio-demographic and attitudinal covariates included in the model (H3c and H3d; but see additional analyses below). Overall, these results offer support for H1c, H1d, H1f, H3a, and H3b, but not for H1a, H1b, H1e, H2a, H2b, H3c, and H3d.

Second, we explored the results of the hypothesis tests in more detail—and regressed each of the four subscale scores on the full set of explanatory variables (see Table 2). These regressions suggest that the *prevalence of the four components of science-related populist attitudes differs substantially across sociodemographic and attitudinal milieus in Switzerland*: Demands for decision-making sovereignty, for example, seem rather loosely related to the explanatory factors included in this study (Adj. $R^2$ = 0.11; $F$ (df) = 4.50 (14, 910), $p$ < 0.001), whereas demands for truth-speaking sovereignty appear to be raised by people who are older, do not hold a university degree, have little proximity to science, and have low trust in scientists (Adj. $R^2$ = 0.30; $F$ (df) = 16.76 (14, 908), $p$ < 0.001; see Table 2). Results also indicate that general perceptions of science are particularly useful in explaining endorsement of the four components, as model fits improved substantially when including these perceptions in stepwise regression models (see Tables A5-A8 in S1 Appendix).

Individual regression estimates provide additional insights: We find that several explanatory variables have similar relationships with multiple components of science-related populist attitudes: For example, lower proximity to science and scientific literacy are associated with more favorable conceptions of the ordinary people and higher demands for decision-making and truth-speaking sovereignty. Age, however, has opposite effects on different components, which perhaps cancel out at the aggregate level of the SciPop Score. Meanwhile, most explanatory variables correlate with just one or two components, such as trust in scientists, which is significantly lower only among people who hold negative conceptions of academic elites and have stronger demands for truth-speaking sovereignty (see Table 2).

Third, we ran two sets of additional analyses to scrutinize the *relationship between science-related populist attitudes and political orientation*. Motivated by research on political populism [54–56], we analyzed if unwillingness or inability to position oneself on the left-right political spectrum is more common among science-related populists. To do so, we repeated the RQ2 regressions but replaced the continuous political orientation measure with three dummy variables that identified left-leaning respondents (scale options 1–3), right-leaning respondents (scale options 5–7), and non-responders ("don't know" or no answer). Moderate respondents (scale option 4) were used as reference group. Results show that non-responders were not significantly more or less likely than responders to endorse science-related populism ($b$ = 0.06, $p$ = 0.727) or any of its components, i.e., conceptions of the ordinary people ($b$ = 0.10, $p$ = 0.586), conceptions of the academic elite ($b$ = 0.21, $p$ = 0.304), demands for decision-making sovereignty ($b$ = 0.24, $p$ = 0.234), or demands for truth-speaking sovereignty ($b$ = 0.27, $p$ = 0.072). After all, there were only few non-responders (6.1%).

RQ2 analyses only tested a monotonic linear association of political orientation and science-related populist attitudes—but are less capable to discern other relationships discussed in the literature, e.g. relationships that are U-shaped [6] or occur only for a specific range of political orientations [59]. We scrutinized these scenarios with two tests: We fitted a quadratic regression model explaining science-related populist attitudes with all explanatory variables used in the RQ2 regression plus the square of political orientation. Results advise against a U-shaped relationship between political orientation and science-related populist attitudes (political orientation × political orientation: $b$ = 0.03, $p$ = 0.266; see Table A9 in S1 Appendix). Afterwards, we performed a two-lines test, which compensates for shortcomings of quadratic regression [85] and has been successfully applied in research on anti-establishment voting and conspiracy beliefs [86, 87]. Results, albeit indicating rather small effects, show that left-leaning

respondents are significantly less likely to hold science-related populist attitudes than moderate respondents ($b = 0.06$, $p = 0.026$), whereas right-leaning and moderate respondents do not differ significantly in their SciPop Scores ($b = 0.08$, $p = 0.078$; see Fig A4 in S1 Appendix). This suggests that science-related populism has a slight tendency to occur on the right and center of the political spectrum—and tends to be less pronounced on the left, which supports H2a.

Fourth, we did two further tests to probe the sensitivity of our analyses: We tested whether we would have obtained different results if we had not applied the Goertz approach to compute the SciPop Score but the Bollen approach—which is theoretically less defensible but still commonly used in populism research [22]. We therefore repeated all hypothesis tests using Bollenian SciPop Scores (mean of all SciPop Scale items) instead of Goertzian SciPop Scores (minimum subscale mean). Bollenian analyses yielded results that did not differ considerably from those of the original analyses: Science-related populist attitudes were still negatively associated with university education, proximity to science, and scientific literacy, for example (see Table A10 in S1 Appendix). Many associations were even stronger than in the Goertzian analyses, which is plausible because Bollenian SciPop Scores tend to be less conservative estimates of science-related populist attitudes than Goertzian SciPop Scores. As a consequence, some explanatory factors showed significant correlations with science-related populism, although they had not done so in the Goertzian analyses (e.g., trust in scientists and religiosity). Meanwhile, Bollenian analyses did not offer clear evidence that science-related populism is less widespread in Italian-speaking Switzerland and more widespread among people with high interest in science, even if the original analyses had showed this (see Table 2).

We also tested if our analyses were sensitive to variance in survey weights, because extreme variance can confound the estimation of regression coefficients [88]. The weights in our data did not have high variance ($SD = 1.44$), but a few were very small or large (range = 0.04 to 23.56). One way to control for these outliers is to constrain the range of weights in a process called trimming. To determine this range, we relied on the interquartile-range method, which is frequently applied in public opinion research [89–91]. When applying this method to our data, we classified all weights smaller than 0.22 and bigger than 4.56 as outliers (see S1 Appendix for more details). We then trimmed the weights to this interval, re-ran the hypothesis tests —and found that the results differed only very marginally from those of the analyses with untrimmed weights (see Table A11 in S1 Appendix). The direction of all coefficients remained the same, their size changed only slightly, and none of the explanatory variables lost or reached significance, except for two: Residing in Italian-speaking Switzerland did not link to science-related populist attitudes anymore, whereas trust in scientists did.

Overall, the results of these two sensitivity tests support those of the original analyses and substantiate their robustness. Moreover, they indicate that science-related populism might also link to lower trust in scientists (H3d). At the same time, they suggest that the association between science-related populist attitudes and interest in science is less pronounced in case the non-compensatoriness premise of science-related populism is ignored (H3a), and that the association between science-related populist attitudes and residence place may be confounded by our sampling design (H1f). These observations should be further explored in future research.

## Discussion

Organized science, its methods and results have recently been challenged by *science-related populism*, which demands that the common sense and will of the "ordinary people," and not allegedly corrupt academic elites, should determine the production of "true knowledge" [6]. Yet there has been no systematic empirical evidence as to how widespread *science-related populist attitudes* are among the population and how they can be explained.

Our study addressed this caveat. It shows that only a small minority of the Swiss support the full spectrum of science-related populism, whereas a large majority do not. But we also find that sizable portions of the population endorse components of science-related populism, which suggests that a considerable number of people may be prone to developing support for the full range of science-related populist ideas, especially if these ideas are articulated in "public arenas" [92] that could foster the success of science-related populist sentiment at the societal level. So far, however, such concerns might be less warranted than those about a rise of political populism in Switzerland [93]: When comparing our results with similar surveys, we find that science-related populist attitudes seem less widespread among the Swiss than political populist attitudes.

Further results suggest that science-related populist attitudes can barely be explained with sociodemographic and attitudinal variables like age, gender, urbanity of residence place, and religiosity. Our main analyses did not unveil significant relationships between science-related populism and (dis)trust in science or scientists either, which supports previous research indicating that the two refer to different phenomena [7]. These findings can be interpreted in at least two ways: First, they suggest that science-related populist attitudes are quite evenly distributed among the Swiss. Second, they may also indicate that these attitudes are contingent upon other individual conceptions, experiences, and imaginations: Science-related populism could be indeed associated with age, religiosity, or trust—but only under certain circumstances, i.e., depending on the scientific topics, actors, and processes respondents thought of when being asked about "scientists," "scientific studies," and other general terms used in the SciPop Scale [7, 52, 64]. From this perspective, science-related populism may be, to some degree, rather specific to the public agendas, institutional configurations, and social contexts in which it emerges.

Yet science-related populists also share certain characteristics: For example, we find that people with no university education, little contact to science, and low scientific literacy display more science-related populism. Part of the reason may be that these people are more prone to perceive commonsensical and intuitional epistemologies as legitimate because they lack familiarity with the scientific epistemology, as such familiarity is usually acquired during university education, more prevalent among those personally involved with science, and a key component of individuals' scientific literacy [82]. This assumption ties in with findings indicating that people who rely on their intuition or gut feelings to discern "truth" are more likely to engage in conspiracist ideation [94].

Results also indicate that science-related populists have a somewhat stronger interest in science. Apparently they exhibit some sort of general cognitive engagement with science—similar to "critically interested" science audience segments in which lower trust and higher attentiveness to science parallel each other [29]. This may seem counterintuitive as it contradicts earlier results [66]. However, existing scholarship suggests that skepticism toward elite institutions (e.g., science) does not necessarily translate into disinterest in their "products" (e.g., scientific knowledge): For example, people who distrust "mainstream media" have been shown to still use them regularly [95].

Additional analyses indicate that supporters of science-related populism do not exhibit "ninisme," i.e., reluctance to self-placement on the left-right spectrum of political orientations [96], but are more likely to identify with right-leaning and moderate orientations and oppose left-leaning orientations. This suggests that left-leaning political ideologies provide fewer affordances for science-related populism [57]. Meanwhile, we do not find that science-related populists favor views at both ends of the political spectrum. This would have concurred with an "extremity hypothesis" that has been discussed in research on (political) populism and posits that populist worldviews often affiliate with radical political ideologies [10, 53]. Neither do we

find support for the assumption that science-related populist attitudes are exclusively linked to right-leaning political views, which would have been along the lines of a "radical-right hypothesis" presuming that right-leaning ideologies provide fertile ground for populist attitudes [10]. Our study rather suggests what could be called a "lenient-left hypothesis" that describes left-leaning citizens as *less* prone to science-related populism than moderate *and* right-leaning citizens. When exploring the different components of science-related populist attitudes, we show that some explanatory factors affect several components simultaneously and in a similar way. However, we find that the explanatory power of these factors varies across components, and that some factors explain only one component. This can be interpreted as evidence for the heterogeneity of public beliefs about science [52]. Moreover, it suggests that the conceptual dimensions of science-related populist attitudes adhere to different psychological mechanisms —which indicates that each dimension adds a unique nuance to science-related populism and underscores that it is "greater than the sum of its parts" [22], i.e., a non-compensatory concept relying on the concurrent presence of all its components [20].

Generally, our findings demonstrate that science-related populism may thrive in segments of the Swiss population and could thus explain increasing fragmentation of science audiences to some degree [97]. But some of these findings require closer scrutiny: Sensitivity tests showed that people's interest in science and place of residence might not be as closely linked to science-related populism as the main analyses indicated, whereas trust in scientists could indeed be. Future studies should therefore try to replicate our own—perhaps also in countries other than Switzerland: This would allow probing if Switzerland, a country with positive public views on science [13] that are usually not challenged by polarized debates [98], provides different "opportunity structures" for science-related populism [99] than countries where perceptions of science are more critical (like Brazil) [5], tied to religious doctrines (like India) [100], or fragmented along partisan lines (like the US) [24].

Follow-up studies should also investigate further potential covariates of science-related populist attitudes, such as income [83], technocratic orientations [32], conspiratorial thinking [101], or intuitive cognitive style [102]. Another such covariate may be anomie—the perception of feeling threatened by the complexity and unpredictability of modern societies [103]—which resonates with endorsement of a "science-related heartland" [6] and was already shown to be stronger among people who hold anti-intellectual attitudes [27] and have low trust in scientific institutions [64]. Studies like these should also scrutinize the association between science-related populism and trust in science or scientists: We modelled trust as an antecedent of populism, drawing on existing research that conceived trust as a rather stable disposition, which may shape more volatile orientations, such as sciences-related populist attitudes [70]. Accordingly, our analyses tested how much variance in science-related populist attitudes can be explained by trust when other covariates remain constant. But science-related populism might also be an antecedent of trust in science [35], so future studies should also test how much variance in trust can be explained by science-related populism. Eventually, these studies should also investigate the temporal stability, context-dependency, and emergence of science-related populism: Panel surveys and experiments could examine volatility over time [104] and differences across topics [9], and qualitative studies may explore how science-related populist attitudes are acquired during socialization [105], triggered by key events [18], and driven by conspiracy theorists like Daniele Ganser in Switzerland or alternative experts like Didier Raoult in France [102].

After all, normative evaluations of our findings are in order. First, such evaluations may address repercussions of science-related populism on science, its democratic legitimization, and the role of expertise within society. Second, they may ask whether science education policies are needed to prevent the development of science-related populist attitudes among

adolescents [106]. Third, they should discuss the implications of science-related populism for science communication, scrutinizing if scientists and practitioners should engage in dialogue with science-related populist audiences and how these might be reached [107]. Some of these discussions, e.g. about the "promises and perils" [108] of "post-normal science communication" [109], have already begun and need to be continued in the future.

## Supporting information

**S1 Appendix Additional information, tables, and figures.**
(PDF)

**S2 Appendix Original questionnaires in German, French, and Italian.**
(PDF)

## Author Contributions

**Conceptualization:** Niels G. Mede, Mike S. Schäfer.

**Formal analysis:** Niels G. Mede.

**Funding acquisition:** Mike S. Schäfer, Julia Metag.

**Methodology:** Niels G. Mede, Mike S. Schäfer, Julia Metag.

**Project administration:** Mike S. Schäfer, Julia Metag.

**Writing – original draft:** Niels G. Mede.

**Writing – review & editing:** Mike S. Schäfer, Julia Metag, Kira Klinger.

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
