## [Decision Letter · Decision Letter 0]

16 Jul 2021

PONE-D-21-17250

How prevalent is science-related populism in Switzerland and what explains it? Evidence from a nationally representative survey

PLOS ONE

Dear Dr. Mede,

Thank you for submitting your manuscript to PLOS ONE. After careful consideration, we feel that it has merit but does not fully meet PLOS ONE’s publication criteria as it currently stands. Therefore, we invite you to submit a revised version of the manuscript that addresses the points raised during the review process.

We look forward to receiving your revised manuscript.

Kind regards,

Christine Mohr, PhD

Academic Editor

PLOS ONE

Additional Editor Comments:

I am delighted that I have received the evaluation of the current manuscript by two experts in the field, with Matteo Cavallaro signing his review. Both have clearly been favorable and gave very clear indications on how to further improve the manuscript. Thus, I do not see any need to further comment on the manuscript and their comments. Rather, I share their support for the current topic, its relevance and study approach.

Journal Requirements:

Reviewers' comments:

Reviewer's Responses to Questions

**Comments to the Author**

1. Is the manuscript technically sound, and do the data support the conclusions?

Reviewer #1: Yes

Reviewer #2: Yes

2. Has the statistical analysis been performed appropriately and rigorously? 

Reviewer #1: Yes

Reviewer #2: Yes

3. Have the authors made all data underlying the findings in their manuscript fully available?

Reviewer #1: Yes

Reviewer #2: Yes

4. Is the manuscript presented in an intelligible fashion and written in standard English?

Reviewer #1: Yes

Reviewer #2: Yes

5. Review Comments to the Author

Reviewer #1: The study presents original and interesting data on a topic potentially central in the year(s) to come, namely science-related populism. It does so by developing a descriptive and exploratory framework which is still suitable for a study in a relatively less touched area. In the literature, populism is increasingly seen as a plural and multi-faceted phenomenon and adding science-related populism is an important element and I welcome the authors’ effort. The article is overall clear, but there are some limitations that should be addressed before it becomes ready for publication.

Structure

- Pages 9/12: Despite being mostly descriptive/exploratory, the article does have an inferential part, as the authors test different hypotheses on the socio-demographic and attitudinal predictors of science-related populism. I think that this part would be easier to grasp for the reader if the authors clearly summarized and expressed their hypotheses. This should be a rather quick modification: pages 9 to 12 are already well structured, what they are missing is just some explicit hypotheses.

- Pages 6/8:As the authors will see, I do have quite a few questions on the conceptual framework. None would require overhauling the article, but there might be a trouble with the word count. My suggestion would be to reduce the space for pages 6 to 8 as they mostly present descriptive data from other studies and re-analyses.

Conceptual Framework

- Pages 8/10/11: I think that the reasoning of the article would be improved by adding a paragraph clarifying the differences between the construct of science-related populism on the one hand and trust in science and scientists. While the authors are honest in recognizing the presence of overlaps between these concepts, they do not provide elements to disentangle the relationship. To put it bluntly: why do they see these variables as independent ones and not as constitutive elements of the dependent variable?

- Pages 9/12 – Sociodemo and political populism: In discussing the sociodemographic predictors of science-related populism, the authors draw a (useful) comparison with the usual determinants of political populism and/or anti-establishment attitudes. In doing so they present different papers that appear to lead to contradictory results (on the urban/rural, gender etc.). However, these results are contradictory mostly because they relate to different forms of political populism: multi-faceted in nature, political populism might have varying socio-demographic expressions. I think that the articles the authors quote on the urban/rural divide speak in favour of such interpretation: Rossi’s article [43] talks about the Italian Five-Star Movement (a form of populism with no labels) while Mamonova and Franquesa deal with right-wing populism (and how to tackle it). And so could (and, given the results, does) science-related populism. Age, gender, urban/rural divide, and political orientation are all not significant or with a small magnitude in their effect. Just as they are contradictory when it comes to political populism taken as a whole (and not divided into left/right-wing populism), so they are not significant when it comes to science-related populism.

As the existence of different forms of political populism with different predictors appears to be quite well established in the literature, the authors should engage with what that means for science-related populism, both in Switzerland and outside of it (I doubt, for example, that political orientation would turn out to be not significant in the US).

- Pages 12 and 17 - Research priorities: as it stands not enough place is given to present the six fields analysed by authors and we find out what these fields are only late in the paper, when looking at the regression tables. I understand the length limits, and as the paper is already charged with other elements, probably it would be best to cut this part: it does not have a proper introduction, the choice of these specific priorities (and relative expectations) is not justified, and results are a bit underwhelming.

Data & Methodology

- Results from both the quadratic model and the two-line test are indeed interesting as they show a lack or a very small effect of self-identification on the LR scale on science-related populism. I wonder, however, if one should not also have a test on those who do not identify on the LR spectrum. Indeed, the ninisme is an existing phenomenon related to different strains of populism, be it on the right (as it was the case for the French FN/RN, see Perrineau) or wwithout labels, as the M5S (see the book by Biorcio or some of the studies by the Italian CISE). So, if it is possible given the data, it is probably worth exploring this relationship as a large minority of populist respondents in other countries (for M5S it was around 16%, for example) do not actually place themselves on the LR scale.

Other minor elements

- Page 6, line 133: I am not certain that the studies presented by authors support the claim that “sizeable portions of many Western populations share beliefs that conceive the ordinary people as homogeneous and virtuous”. While the claim is supported for the virtuous part, the re-analysis of Spruyt, Keppens, and van Droogenbroeck’s study (at least as it is presented) refers to a feeling of connection which seems to me more related to a form of people identification rather than an epistemological one.

- Page 9: I am not certain that Rico et al. article is the best in supporting the idea that losers of globalization feel threatened by scientific innovations (and, thus, turn to populism). Their main conclusion was that (semi-objective) vulnerability was a worse predictor than the subjective judgement on the national economy. Even if it is article specific to right-wing populism (and party’s preferences rather than populist attitudes), I’d suggest to switch to Im, Mayer, Palier, Rovny 2019, which tackles more this aspect of the losers of automation (and not just globalization). You could also refer to Rydgren 2007 and the idea of losers of modernization (rather than just globalization).

- Page 10, lines 225-226: is there a reason to expect an effect of the cantonal language? If studies have contradictory results and there is no theoretical expectation, there could simply be presented as a control for the questionnaire formulation. If there is a valid reason to expect diverging results, then this should be presented.

- Page 34, table 1: I would add as a note the alpha score for the mean across al items of both science-related and political populist attitudes. While not so relevant in a Goertz approach, alphas are quite informative when averaging items.

- Page 34, table 1, note c: I think that the best argument in favor of not considering the CFA is the non-compensatory approach, rather than the existence of sub-dimensions within the dimension. If it was just a matter of sub-dimensions, this could easily be solved by either adding the four sub-constructs or playing with the error covariances. If, however, the dimension is not compensatory, factor analysis as a whole is not so useful.

- In the data description, add the standard deviation, maximum and minimum of the weights.

Reviewer #2: The paper entitled "How prevalent is science-related populism in Switzerland and what explains it? Evidence from a nationally representative survey" aims at investigating the prevalence of a special case of populist attitudes, namely science-related populist attitudes, and some of its potential predictors. For this aim, the authors analysed a representative sample of the swiss population.

This study seems to me very timely and interesting, because after more that one year of pandemic, we all observed the rise of anti-science attitudes, and the social troubles they created, up to some damages to democracy itself. Regarding its scientific value, the study is well conducted and analysed, the results are strong given the representativeness of the sample, and could in my view be published with minor revisions, listed below, in PLOS ONE.

1) P. 2: In the first line of the summary, the "distinct" qualifyer of science-related populism should be avoided, as there is no empirical evidence in order to support such claim (a general populism scale should have been added in the study, as in Castanho Silva et al., 2017). So I would recommend to write more generally "a variant of populism". In the limitations of the study, it should be added that the distinctiveness of the science-related populism compared to the general political populism has still to be empirically tested.

2) A word could be added in the introduction section about the special case of minority (even renowned) scientists such as Didier Raoult in France, who fostered and still fosters conspiracy theories and populist attitudes, which are very popular in France (hundreds of thousands of views on Youtube). His narcissism (to think being right against the whole scientific community), populism (he derogates the "methodologists" and "big data analysts" but values "real doctors" like him who directly heal patients, etc. ; but as the same time presents himself as the "real elite" against the other mediocre scientists) and adherence to conspiracy theories (pharmaceutical companies drowned hydroxychloroquine in order to sell vaccines) really fuel science-related populism (see Bertin et al., 2021; Fuhrer & Cova, 2020).

3) P. 13: The “Goertz approach” seems to have some merit to get a non-compensatory measure, but it seems to add some confusion in the results, some participants having a SciPop score about perception of ordinary people, some others about academic elites, and other about sovereignty. In my view, this sounds like adding apples and oranges in the same score. So I would recommend the authors to report the "Goertz" and the classical "Bollen" scores when commenting the general level of science-related populism in the population, but to compute and report the classical Bollen score in the regression analyses. In these analyses, the fact that the different participants may have various subscale scores as a general score really seem to bring confusion in the results.

Relatedly, in pp. 15-16, there is a repetition of the mention of a SciPop Score of 2.22. It would be clearer to have only once the overall mean reported.

The Cronbach alpha of the whole scale should be reported, as well as the bivariate correlations between the pairs of agglomerated items (the whole correlation table with all variables should be reported in the appendix), in order to ascertain the factorial structure of the whole scale and subscales. This is important because the study rely on these measures.

References :

Bertin, P., Nera, K. & Delouvée, S. (2020). Conspiracy beliefs, rejection of vaccination, and support for hydroxychloroquine: A conceptual replication‐extension in the Covid‐19 pandemic context. Front. Psychol. 11:565128. En ligne : https://doi.org/10.3389/ fpsyg.2020.565128

Castanho Silva, B., Vegetti, F. & Littvay, L. (2017). The elite is up to something: Exploring the relation between populism and belief in conspiracy theories. Swiss Political Science Review, 23, 423‐443.

Fuhrer, J. & Cova, F. (2020). “Quick and dirty”: Intuitive cognitive style predicts trust in Didier Raoult and his hydroxy‐ chloroquine‐based treatment against COVID‐19. Judgement and Decision Making, 15, 889‐908.

6. PLOS authors have the option to publish the peer review history of their article (what does this mean?). If published, this will include your full peer review and any attached files.

Reviewer #1: **Yes: **Matteo Cavallaro

Reviewer #2: No

---

## [Author Response · Author response to Decision Letter 0]

28 Aug 2021

Response to Reviewer Comments on ms. “How Prevalent Is Science-Related Populism in Switzerland and What Explains It? Evidence from a Nationally Representative Survey”

We thank the reviewers for their helpful comments on the manuscript and are glad that they think it presents relevant findings which advance scholarship on a timely topic. Their recommendations pointed us to weaknesses we had not recognized before and contained important suggestions on how to improve the paper. We were happy to follow these recommendations and edited the manuscript accordingly, as we will detail below. We hope that the editor and reviewers find it publishable in its revised form.

Reviewer 1 (R1)

R1: Pages 12 and 17: Research priorities: As it stands, not enough place is given to present the six fields analyzed by authors and we find out what these fields are only late in the paper […]. […] [P]robably it would be best to cut this part: it does not have a proper introduction, the choice of these specific priorities (and relative expectations) is not justified, and results are a bit underwhelming.

AUTHORS: We respond to this comment first, because addressing it had implications for multiple parts of the paper. We agree with these concerns—the analyses of research priority perceptions might have lacked a comprehensive embedding in the article and may not add very much to its overall merit. Therefore, we excluded these analyses from the study and removed all parts where they are mentioned (literature review, hypotheses, method section, results, discussion). This also changed the results of all regressions and additional analyses very slightly—on the one hand because of correlations between priority perceptions and other explanatory factors, and on the hand because the new analyses now include a handful of additional respondents who had previously produced missing cases as they did not respond to one or more of the six variables. Importantly, the new results of the hypothesis tests differ only very marginally from the previous: Mostly, only the last digits of the regression estimates, standard errors, and p values changed. None of the explanatory variables of science-related populist attitudes reached or lost significance, so the findings remained essentially the same.

R1: Pages 6/8: Reduce the space for pages 6 to 8 as they mostly present descriptive data from other studies and re-analyses.

AUTHORS: Good recommendation, we did that.

R1: Pages 9/12: Despite being mostly descriptive/exploratory, the article does have an inferential part […]. I think that this part would be easier to grasp for the reader if the authors clearly summarized and expressed their hypotheses. This should be a rather quick modification: pages 9 to 12 are already well structured, what they are missing is just some explicit hypotheses.

AUTHORS: This is a very good idea, so we have now included three sets of hypotheses which assume associations of the science-related populist attitudes and sociodemographic characteristics (H1a-f), political orientation and religiosity (H2a-b), and general perceptions of science (H3a-d). The ambiguousness of existing conceptual and empirical scholarship did not allow us to hypothesize about the direction of these associations, though: Such scholarship seems indeed to be quite context-dependent (see below) and suggests that critical attitudes toward science vary across topics, institutions, cultures, and countries—with very few evidence on Switzerland, and no systematic evidence on science-related populist attitudes in particular. Therefore, we formulated non-directional hypotheses and included explanations of why we did so in the text.

R1: Pages 8/10/11: I think that the reasoning of the article would be improved by adding a paragraph clarifying the differences between the construct of science-related populism on the one hand and trust in science and scientists. While the authors are honest in recognizing the presence of overlaps between these concepts, they do not provide elements to disentangle the relationship. To put it bluntly: why do they see these variables as independent ones and not as constitutive elements of the dependent variable?

AUTHORS: You raise an important point which indeed needs more explanation. Therefore, we added conceptual and empirical arguments to disentangle the difference between science-related populism and (dis)trust in science: Conceptually, the two differ in that science-related populism implies demands of the “ordinary people” to participate in science—which are typically not conceived as an essential element of distrust toward science. Empirically, trust in science/scientists and science-related populist attitudes correlate only moderately (Mede, Schäfer, & Füchslin, 2021), which suggests that they do have overlaps but are also different from each other. Accordingly, (dis)trust in science constitutes indeed single aspects of science-related populism—which is why we assume an association between the two—but (dis)trust contains only single aspects of it, which is why it is also partly independent from science-related populism. We have now included these arguments in the beginning of the conceptual part.

Meanwhile, we sensed that the second part of the comment (“why do they see these variables as independent ones…?”) may indicate a further lack of clarity: We do not assume a causal relationship between trust in science and science-related populist attitudes, with the former being an independent predictor and the latter a dependent variable. Some formulations in the first version of the manuscript might have implied this assumption—so we replaced these (e.g., predict, affect, influence, etc.) with non-causal formulations (e.g., association, relationship, correlate, link, etc.).

R1: Pages 9/12: […] [M]ulti-faceted in nature, political populism might have varying socio-demographic expressions. I think that the articles the authors quote on the urban/rural divide speak in favour of such interpretation: Rossi’s article [43] talks about the Italian Five-Star Movement (a form of populism with no labels) while Mamonova and Franquesa deal with right-wing populism (and how to tackle it). And so could (and, given the results, does) science-related populism. Age, gender, urban/rural divide, and political orientation are all not significant or with a small magnitude in their effect. Just as they are contradictory when it comes to political populism taken as a whole (and not divided into left/right-wing populism), so they are not significant when it comes to science-related populism.

As the existence of different forms of political populism with different predictors appears to be quite well established in the literature, the authors should engage with what that means for science-related populism, both in Switzerland and outside of it (I doubt, for example, that political orientation would turn out to be not significant in the US).

AUTHORS: Thank you very much for this important comment! You are completely right that populist phenomena—as well as critical, skeptical, and hostile public perceptions of science—and their predictors may vary substantially across contexts, which indeed explains part of the ambiguity of results of existing scholarship and may likely also have consequences for the study of science-related populism. In fact, such variability may not only affect scholarship of sociodemographic predictors but also of attitudinal correlates (i.e. political orientation, religiosity, trust in science, etc.).

The revised manuscript accounts for these points in several places: Firstly, we added some sentences to each of the chapters of the section on “Science-Related Populist Attitudes and Explanatory Factors”. They discuss the heterogeneity of populist and anti-scientific sentiments and their sociodemographic and attitudinal predictors, as well as implications of this heterogeneity for our hypotheses. Secondly, we have now considered the contextual variability of such sentiments as a potential explanation for why we found only few significant associations between science-related populism and sociodemographic (age, gender, etc.) and attitudinal (religiosity, trust, etc.) variables. In particular, we contend that science-related populism may be similarly context-dependent like political populism, for example—and revised the discussion section accordingly. Thirdly, we now reflect on what such context-dependency means for future research on science-related populism, which also led to some edits of the discussion section.

R1: Results from both the quadratic model and the two-line test are indeed interesting as they show a lack or a very small effect of self-identification on the LR scale on science-related populism. I wonder, however, if one should not also have a test on those who do not identify on the LR spectrum. Indeed, the ninisme is an existing phenomenon related to different strains of populism, be it on the right (as it was the case for the French FN/RN, see Perrineau) or without labels, as the M5S (see the book by Biorcio or some of the studies by the Italian CISE). So, if it is possible given the data, it is probably worth exploring this relationship as a large minority of populist respondents in other countries (for M5S it was around 16%, for example) do not actually place themselves on the LR scale.

AUTHORS: That is a very interesting idea, thank you for bringing it up! Our paper now scrutinizes the possibility that science-related populists may be similarly reluctant to identify with left- or right-leaning positions like FN/RN or M5S voters, for example. Hence, we (1) referenced studies of Biorcio and Perrineau in the Literature Review section; (2) ran additional analyses according to your suggestions (i.e., we included all 6.1% respondents who did not respond to the political orientation item or answered “don’t know” in the analyses using dummy variables); (3) included and discussed the results of these analyses (no significant differences between responders and non-responders).

R1: Page 6, line 133: I am not certain that the studies presented by authors support the claim that “sizeable portions of many Western populations share beliefs that conceive the ordinary people as homogeneous and virtuous”. While the claim is supported for the virtuous part, the re-analysis of Spruyt, Keppens, and van Droogenbroeck’s study (at least as it is presented) refers to a feeling of connection which seems to me more related to a form of people identification rather than an epistemological one.

AUTHORS: We understand your concern: Indeed, Spruyt, Keppens, and van Droogenbroeck (2016) argue and show that an “education-based identity” fosters some sort of people identification. However, we think that their argument still has an epistemological component, as it suggests that people perceive the knowledge, expertise, and competences they acquired during education as a common denominator. We hope that we have clarified this in the revised version of the manuscript, in which we removed the result of the re-analysis and referenced Spruyt et al.’s general argument instead. Importantly, we also added a reference to a recent survey in Austria, which indicates that quite some people perceive trust in common sense as a principle which unifies the ordinary people (Eberl, Greussing, Huber, & Mede, 2021). We believe that these two revisions support our claim that “sizeable portions of many Western populations” hold “epistemological […] homogeneity perceptions.”

R1: Page 9: I am not certain that Rico et al. article is the best in supporting the idea that losers of globalization feel threatened by scientific innovations (and, thus, turn to populism). […] I’d suggest to switch to Im, Mayer, Palier, Rovny 2019, which tackles more this aspect of the losers of automation (and not just globalization). You could also refer to Rydgren 2007 and the idea of losers of modernization (rather than just globalization).

AUTHORS: A very helpful suggestion, thank you! We changed this part accordingly and now refer to Im et al. (2019).

R1: Page 10, lines 225-226: is there a reason to expect an effect of the cantonal language? […] If there is a valid reason to expect diverging results, then this should be presented.

AUTHORS: Yes, there is reason to expect such differences: Several conceptual and empirical studies have described cultural differences between the linguistic regions of Switzerland and showed that these manifested in diverging political and science-related attitudes, for example. This led us to assume that science-related populist attitudes may also differ across linguistic regions. However, we had not made this clear enough in the original version of the manuscript—and have now added some sentences in the chapter on sociodemographic predictors of science-related populism.

R1: Page 34, table 1: I would add as a note the alpha score for the mean across all items of both science-related and political populist attitudes. While not so relevant in a Goertz approach, alphas are quite informative when averaging items.

AUTHORS: Done.

R1: Page 34, table 1, note c: I think that the best argument in favor of not considering the CFA is the non-compensatory approach, rather than the existence of sub-dimensions within the dimension. If it was just a matter of sub-dimensions, this could easily be solved by either adding the four sub-constructs or playing with the error covariances. If, however, the dimension is not compensatory, factor analysis as a whole is not so useful.

AUTHORS: Good point—we revised the note according to your suggestion.

R1: In the data description, add the standard deviation, maximum and minimum of the weights.

AUTHORS: Done.

Reviewer 2 (R2)

R2: In the first line of the summary, the “distinct” qualifier of science-related populism should be avoided, as there is no empirical evidence in order to support such claim (a general populism scale should have been added in the study, as in Castanho Silva et al., 2017). So I would recommend to write more generally “a variant of populism”. In the limitations of the study, it should be added that the distinctiveness of the science-related populism compared to the general political populism has still to be empirically tested.

AUTHORS: We agree with your comment—claiming that science-related and political populism are generally distinct is not warranted in absence of empirical evidence. Therefore, we changed the formulation to “a specific variant of populism” (or removed it in the Discussion section). We also added some sentences in the beginning the conceptual part which explain that such specificity is due to the conceptual distinctiveness of science-related populism and political populism (e.g., because the former refers to “scientific instead of political authorities” and raises “scientific truth claims instead of political power claims”). Mede and Schäfer (2020), whose work we refer to, provide a more detailed account of this argument. Nevertheless, the empirical distinctiveness of political and science-related populism still needs to be investigated—which we have now pointed out in the beginning of the Discussions section.

R2: A word could be added in the introduction section about the special case of minority (even renowned) scientists such as Didier Raoult in France, who fostered and still fosters conspiracy theories and populist attitudes, which are very popular in France (hundreds of thousands of views on Youtube). His narcissism (to think being right against the whole scientific community), populism (he derogates the “methodologists” and “big data analysts” but values “real doctors” like him who directly heal patients, etc. ; but as the same time presents himself as the “real elite” against the other mediocre scientists) and adherence to conspiracy theories (pharmaceutical companies drowned hydroxychloroquine in order to sell vaccines) really fuel science-related populism (see Bertin et al., 2021; Fuhrer & Cova, 2020).

AUTHORS: Thank you for these valuable suggestions! We are now referring to Raoult as an example for a “pseudo-expert”—but in the Discussion section instead of the Introduction, because we felt the first paragraphs of the article should lead the reader as quickly as possible to science-related populist attitudes (the “demand side” of science-related populism; see Mede & Schäfer, 2020) instead of potential science-related populist communicators (the “supply side” of science-related populism).

Moreover, your comment was also helpful in pointing us to the paper by Fuhrer and Cova (2020), which suggests that an intuitive cognitive style may be a correlate of science-related populist attitudes and as such could be studied in further research, as we now point out in the Discussion section.

R2: P. 13: The “Goertz approach” seems to have some merit to get a non-compensatory measure, but it seems to add some confusion in the results, some participants having a SciPop score about perception of ordinary people, some others about academic elites, and other about sovereignty. In my view, this sounds like adding apples and oranges in the same score. So I would recommend the authors to report the “Goertz” and the classical “Bollen” scores when commenting the general level of science-related populism in the population, but to compute and report the classical Bollen score in the regression analyses. In these analyses, the fact that the different participants may have various subscale scores as a general score really seem to bring confusion in the results.

AUTHORS: We understand your concerns about the Goertz approach, which actually echo current debates within scholarship of political populism. We are nevertheless confident that this approach represents the most useful way to quantify science-related populist attitudes and explain them in regression analyses, whereas Bollen and Sartori approaches are less defensible. However, the manuscript was perhaps not entirely clear about this. Therefore, we have taken some effort to add more detailed arguments explaining that the Goertz approach can be understood as the only valid way to operationalize multi-dimensional non-compensatory phenomena like science-relate populism, which can only exist if all dimensions are present (Mede et al., 2021; Wuttke, Schimpf, & Schoen, 2020). 

This approach may indeed seem like adding apples (e.g., conceptions of the ordinary people), oranges (e.g., conceptions of the academic elite), and perhaps bananas (e.g., demands for decision-making sovereignty), and berries (e.g., demands for truth-speaking sovereignty). But it is precisely the mix of all these four fruits that constitutes science-related populism, metaphorically speaking. The Bollen approach, however, would also sell a mix that contains many apples and oranges but no bananas and berries as science-related populism—even though it really isn’t, because we need bananas and berries for the full flavor of science-related populism. This is why we would consider Bollenian SciPop Scores as not appropriate—both when reporting the overall level of science-related populism and investigating explanatory factors in the regression analyses—and prefer the Goertzian SciPop Score instead.

Moreover, our procedure is consistent with an emerging standard of research on political populism that increasingly uses Goertzian scores as well, which we now also point out in the manuscript. In addition, we now use more consistent wording to refer to the different aggregation approaches in the main text and the tables in order to improve understandability. We hope very much that these edits clarify how and why we chose the Goertz approach throughout all analyses, and remove confusion from the results.

R2: Relatedly, in pp. 15-16, there is a repetition of the mention of a SciPop Score of 2.22. It would be clearer to have only once the overall mean reported.

AUTHORS: Thanks for pointing that out, we changed that and removed the repetition. The overall mean (i.e. the Bollenian SciPop Score), which we prefer not to mention prominently in the main text for the reasons mentioned above, can be found in Table 1 (i.e., it is 3.02).

 

References

Eberl, J.‑M., Greussing, E., Huber, R. A., & Mede, N. G. (2021). Wissenschaftsbezogener Populismus: Eine österreichische Bestandsaufnahme: [Science-related populism in Austria: Taking stock]. Retrieved from https://viecer.univie.ac.at/corona-blog/corona-blog-beitraege/blog124/

Fuhrer, J., & Cova, F. (2020). “Quick and dirty”: Intuitive cognitive style predicts trust in Didier Raoult and his hydroxychloroquine-based treatment against COVID-19. Judgment and Decision Making, 15(6), 889–908.

Mede, N. G., & Schäfer, M. S. (2020). Science-related populism: Conceptualizing populist demands toward science. Public Understanding of Science, 29(5), 473–491. https://doi.org/10.1177/0963662520924259

Mede, N. G., Schäfer, M. S., & Füchslin, T. (2021). The SciPop Scale for measuring science-related populist attitudes in surveys: Development, test, and validation. International Journal of Public Opinion Research, 33(2), 273–293. https://doi.org/10.1093/ijpor/edaa026

Spruyt, B., Keppens, G., & van Droogenbroeck, F. (2016). Who supports populism and what attracts people to it? Political Research Quarterly, 69(2), 335–346. https://doi.org/10.1177/1065912916639138

Wuttke, A., Schimpf, C., & Schoen, H. (2020). When the whole is greater than the sum of its parts: On the conceptualization and measurement of populist attitudes and other multidimensional constructs. American Political Science Review, 114(2), 356–374. https://doi.org/10.1017/S0003055419000807

---

## [Decision Letter · Decision Letter 1]

4 Jan 2022

PONE-D-21-17250R1How prevalent is science-related populism in Switzerland and what explains it? Evidence from a nationally representative surveyPLOS ONE

Dear Dr. Mede,

Thank you for submitting your manuscript to PLOS ONE. I am pleased to inform you that both referees were very appreciative of the first revision of your manuscript. There are some minor issues they wish you to consider, mainly with respect to data treatment. Both referee were very detailed in their suggestions, as you can extract from their reports. Please consider their suggestions carefully.

Accordingly, we feel that your manuscript has merit but does not fully meet PLOS ONE’s publication criteria as it currently stands. Therefore, we invite you to submit a revised version of the manuscript that addresses the points raised by the two referees. 

We look forward to receiving your revised manuscript.

Kind regards,

Christine Mohr, PhD

Academic Editor

PLOS ONE

Journal Requirements:

Reviewers' comments:

Reviewer's Responses to Questions

**Comments to the Author**

1. If the authors have adequately addressed your comments raised in a previous round of review and you feel that this manuscript is now acceptable for publication, you may indicate that here to bypass the “Comments to the Author” section, enter your conflict of interest statement in the “Confidential to Editor” section, and submit your "Accept" recommendation.

Reviewer #1: All comments have been addressed

Reviewer #2: (No Response)

2. Is the manuscript technically sound, and do the data support the conclusions?

Reviewer #1: (No Response)

Reviewer #2: Partly

3. Has the statistical analysis been performed appropriately and rigorously? 

Reviewer #1: Yes

Reviewer #2: No

4. Have the authors made all data underlying the findings in their manuscript fully available?

Reviewer #1: Yes

Reviewer #2: Yes

5. Is the manuscript presented in an intelligible fashion and written in standard English?

Reviewer #1: Yes

Reviewer #2: Yes

6. Review Comments to the Author

Reviewer #1: I wish to thank the authors for the good work they have carried out so far and their complete and polite replies to my comments. I think that the article has clearly improved and we are close to a definitive and publishable version. I still have, however, two doubts:

1. Weights: to the best of my knowledge, it is commonly considered a best practice to trim excessive weight to avoid the possible effects of outlying weigthed observations. A commonly applied trimming policy is at a around 5 or 7 max (and usually 1/5 or 1/7 as a min weight). Having 0.04 as min and close to 24 as max does rise some concerns on the possible effect of outlying weigthed observations. As such, I would add, as a robustness check, a rerun of Table 2 with trimmed weights.

2. Trust in science/scientists as independent variables: This is more a suggestion than a request like the one before. There is a certain tendency in contemporary literature to, let me be blunt, run exploratory analyses with inferential models (such as the OLS). I understand that the authors do not claim causality, still in a multivariate OLS you are testing a model where you expect a certain directionality. This is actually still present in the text: these are your explanatory variables (and it is clearly expressed in the text, in the table and in the R code). Why do expect these Xs to explain your dependent variable? What is the theoretical assumption behind the fact that you decided to put them on the right side instead of the left side of the equation? Do not get me wrong: this changes nothing to the rest of the article, but you are not just showing a correlation matrix identifying the significant corr between variables. By using a regression you are testing a series of hypotheses, in this case of an addictive effect of trust in science/scientists keeping all other covariates equal. You cannot claim causality and say "this is the correct directionality", but you still can and should say why, theoretically, you think this is the proper way to represent and test this relationship. While true in general, in this case given the clear conceptual overlap between the constructs it would be better to clarify the authors' reasoning: just adding covariates because they turned out to be related/significant in other studies is not enough. A short paragraph, given the large amount of other variables that have been tested, would suffice and improve the overall quality of the text, while at the same time recognizing a limit and setting up new research (that I would gladly read by the way).

Reviewer #2: I thank the authors for improving the quality of the paper after this first step of revision. Unfortunately, I still have a serious concern about the computation of the science-related populism score. Less importantly, the specificity of science-related populism should be better underlined (or dimmed).

– About the "goertzian" vs. "bollenian" approaches of computing the populism score, the explanations of the authors lend me to think that, contrary to their intention, the goertzian appears to me as a bad response to a good question. The justification of the authors for choosing the goertzian approach is the following :

"Averaging participants’ responses to the SciPop Scale items would have been another way to compose an aggregate score for science-related populist attitudes—a procedure Wuttke, Schimpf, and Schoen [5] described as “Bollen approach.” We decided against this approach, because it would produce scores which are not in line with the conceptual premise that science-related populism is non-compensatory, i.e. that it requires the concurrent presence of all its four components [6]. For example, Bollen scores (means of all scale items) would indicate similar degrees of science-related populism for an individual A who endorses some of its components fully but rejects others completely and for an individual B who endorses all components moderately—although only people like B can be conceptualized as supporters of science-related populism as all its facets are concurrently present in them. Goertz scores (minimum subscale means) would not indicate similar populism degrees for A and B, however. They would be small in A’s case, where one component is absent, and bigger in B’s case, where all components are present. Accordingly, the Goertz approach can usefully account for the concurrency criterion of non-compensatory phenomena such as science-related populism, and, as such, represents a useful analytical procedure for us to translate responses to the SciPop Scale into a single numerical value indicating “full-fledged” science-related populist attitudes." (Appendix S1).

I am not convinced by this justification. If the authors fear that some individuals could endorse some aspects and reject other aspects of scientific populism, and that scientific populism should be non-compensatory, my opinion is that they solve this problem by the goertzian approach by adding more serious problems. In regression analyses, the authors will predict a score that is not similar across participants! As the score does not reflect the same items, I really do think that the regression will be statistically flawed (because some of the predictors of the same analysis will predict some items for some participants, and some other items for other participants).

The crucial question whether subdimensions are correlated or not may be in my view more classically and properly (I am not opposed of course by new approaches and analyses, but not if they seem to unclarify the analyses, as it appears to me here) addressed by correlational analyses (Cronbach's alpha, and/or oblique PCA, and/or model comparisons of 1-factor vs. 4-factors solutions by CFA). Perhaps the simplest and closest to the existing results way of doing this is to perform a regression on the overall average score with a sufficient Cronbach alpha, and then performing distinct regressions for each subscore, as still presented in table 2 (so only changing the goertzian score by a bollenian score in the overall regression).

I would really (and I guess other scholars also will) be more confident about the results of the regression on the classical average score (Bollen), rather than on the goertzian score.

I would also add that the authors used the bollenian approach when computing the four subdimensions of populism (average of the two items) and not the goertzian approach (lower score of the two items), which is not consistent. But the computation of the goertzian score may be kept for comparison with other studies (in table 1).

– About my remark on the specificity of science-related populism, I am not fully satisfied by the response of the authors. I would remove the term "specific" and let only "variant" in the summary and along the article, as I still advocated for. For me, replacing "distinct" by "specific" is only some kind of rephrasing. Similarly, the sentence p. 5 "Such anti-scientific ideas represent a specific variant of populism, which differs conceptually from political populism—and has been conceptualized as “science-related populism” [6]." is not satisfying to me. Either the authors should explain what is exactly specific with regard to other forms of populism, or they should only write ""Such anti-scientific ideas represent a variant of populism, which has been conceptualized as “science-related populism” [6].". These changes will be more consistent with the conclusion of the authors that "further studies measuring both science-related and political populism must investigate this finding—and could then also test the conceptual assumption that the former and the latter are distinct populism variants".

Minor comments:

– In p. 17, I would propose to say that there is a slight tendency to have more science-related populism on the right that on the left, because all betas are positive, the overall beta is very close to significance, and one beta is significant.

– P. 18, I would precise what groups the authors refer to when saying: "While demands for truth-speaking sovereignty appear to be raised by very specific population groups of the population".

– PP. 19-20, all p-values could be rounded at 3 decimals

– P. 24, I would say that Didier Raoult is an "isolated expert" rather than "pseudo-expert", as he was a renowned expert before covid-19 (and incidentally, you will avoid to be fired on French social media !!!).

7. PLOS authors have the option to publish the peer review history of their article (what does this mean?). If published, this will include your full peer review and any attached files.

Reviewer #1: **Yes: **Matteo Cavallaro

Reviewer #2: **Yes: **Pascal Wagner-Egger

---

## [Author Response · Author response to Decision Letter 1]

10 Mar 2022

Please see Response to Reviewers letter attached to this resubmission.

---

## [Decision Letter · Decision Letter 2]

27 Jun 2022

Who supports science-related populism? A nationally representative survey on the prevalence and explanatory factors of populist attitudes toward science in Switzerland

PONE-D-21-17250R2

Dear Dr. Mede,

We’re pleased to inform you that your manuscript has been judged scientifically suitable for publication and will be formally accepted for publication once it meets all outstanding technical requirements.

Kind regards,

Ghaffar Ali, PhD

Academic Editor

PLOS ONE

Additional Editor Comments (optional):

Reviewers' comments:

Reviewer's Responses to Questions

**Comments to the Author**

1. If the authors have adequately addressed your comments raised in a previous round of review and you feel that this manuscript is now acceptable for publication, you may indicate that here to bypass the “Comments to the Author” section, enter your conflict of interest statement in the “Confidential to Editor” section, and submit your "Accept" recommendation.

Reviewer #1: All comments have been addressed

2. Is the manuscript technically sound, and do the data support the conclusions?

Reviewer #1: Yes

3. Has the statistical analysis been performed appropriately and rigorously? 

Reviewer #1: Yes

4. Have the authors made all data underlying the findings in their manuscript fully available?

Reviewer #1: No

5. Is the manuscript presented in an intelligible fashion and written in standard English?

Reviewer #1: Yes

6. Review Comments to the Author

Reviewer #1: The overall quality of the text seems to me to have clearly improved. The authors replied to all my doubts and even when they (legitimately) disagreed with my suggestions, they were honest enough to add 'a warning' in the text, recognizing the limitations of their study. As such, readers have now access to all relevant information to understand the limits (and the pros) of this research, and the various robustness checks show that the core argument of the research is solid enough for publication.

7. PLOS authors have the option to publish the peer review history of their article (what does this mean?). If published, this will include your full peer review and any attached files.

Reviewer #1: **Yes: **Matteo Cavallaro

---

## [Editor Report · Acceptance letter]

28 Jul 2022

PONE-D-21-17250R2 

Who Supports Science-Related Populism?
A Nationally Representative Survey on the Prevalence and Explanatory Factors of Populist Attitudes Toward Science in Switzerland 

Dear Dr. Mede:

I'm pleased to inform you that your manuscript has been deemed suitable for publication in PLOS ONE. Congratulations! Your manuscript is now with our production department. 

Kind regards, 

on behalf of

Prof. Ghaffar Ali 

Academic Editor

PLOS ONE